# Narrative Review of Biological Markers in Chronic Limb-Threatening Ischemia

**DOI:** 10.3390/biomedicines12040798

**Published:** 2024-04-03

**Authors:** Alexandra Ioana Popescu, Andreea Luciana Rata, Sorin Barac, Roxana Popescu, Roxana Ramona Onofrei, Cristian Vlad, Daliborca Vlad

**Affiliations:** 1Pharmacology Department, Doctoral School, “Victor Babes” University of Medicine and Pharmacy, 300041 Timisoara, Romania; alexandra_popescu2007@yahoo.com; 2Surgical Emergencies Department, “Victor Babes” University of Medicine and Pharmacy, 300041 Timisoara, Romania; rataandreealuciana@gmail.com; 3Vascular Surgery Department, “Victor Babes” University of Medicine and Pharmacy, 300041 Timisoara, Romania; 4Cell and Molecular Biology Department, ”Victor Babes” University of Medicine and Pharmacy, 300041 Timisoara, Romania; popescu.roxana@umft.ro; 5Department of Rehabilitation, Physical Medicine and Rheumatology, Research Center for Assessment of Human Motion, Functionality and Disability, ”Victor Babes” University of Medicine and Pharmacy, 300041 Timisoara, Romania; onofrei.roxana@umft.ro; 6Pharmacology Department, “Victor Babes” University of Medicine and Pharmacy, 300041 Timisoara, Romania; vlad.cristian@umft.ro (C.V.); vlad.daliborca@umft.ro (D.V.)

**Keywords:** chronic limb-threatening ischemia, biological markers, inflammatory markers, endothelial dysfunction

## Abstract

Background: Chronic limb-threatening ischemia (CLTI), the advanced stage of peripheral arterial disease, is diagnosed in the presence of ischemic rest pain, non-healing ulcers, or gangrene. Several studies have demonstrated that inflammation and endothelial dysfunction are some of the main substrates of CLTI. Methods: A narrative review was conducted and reported according to PRISMA guidelines. Three databases were searched—Web of Science, Medline, and EMBASE—for the studies assessing CLTI and the biological markers related to it. Results: We included 22 studies, and all the markers identified (C-reactive protein, D-dimers, fibrinogen, cytokines, IL-6, TNF-α, ICAM-1 (Intracellular Adhesion Molecule-1), VCAM-1 (Vascular Cell Adhesion Molecule-1), neutrophile-to-lymphocytes ratio (NLR), IL-8, Pentraxin-3, neutrophil gelatinase-associated lipocalin (NGAL), calprotectin, E-selectin, P-selectin, neopterin, High-Mobility Group Box-1 protein (HGMB-1), Osteoprotegerin (OPG) and Sortilin) were positively associated with advanced CLTI, with major limb or major cardiovascular events in these patients. Conclusions: All the studied markers had increased values in patients with CLTI, especially when associated with diabetes mellitus, proving a very important association between diabetes and major limb or cardiovascular events in these patients. There is a need for more studies to validate these markers in terms of diagnosis or prognosis in CLTI patients and in trying to find new medical strategies that target inflammation or endothelial dysfunction in these patients.

## 1. Introduction

Peripheral arterial disease (PAD) incidence has increased significantly in the last few years. Most patients are asymptomatic for long periods of time, but at least 10% will progress toward chronic limb-threatening ischemia (CLTI) or will present per primam in this stage of the disease. The modifiable risk factors for PAD and CLTI are smoking, diabetes mellitus, arterial hypertension, chronic kidney disease, sedentary lifestyle, and obesity. Despite the best medical treatment and the progress made in correcting the risk factors, CLTI is still associated with higher morbidity and mortality. Untreated, CLTI leads to an approximately 25% major amputation rate 1 year after the diagnosis [1].

CLTI guidelines published in 2019 [2] proposed that the definition of CLTI should include a broader spectrum and a more heterogenous group of patients, with different ischemia degrees and a higher major amputation risk.

Ischemic rest pain is usually described as affecting the forefoot and is usually aggravated by the decubitus position; for the diagnosis, the pain should be present for more than 2 weeks and associated with one or more abnormal parameters: ankle–brachial index < 0.4, ankle pressure < 50 mmHg in absolute value, hallux pressure under 30 mmHg in absolute value, a transcutaneous oxygen pressure under 30 mmHg, and a flat or weakly pulsating wave curve (equivalent to a third degree of ischemia in the WIfI classification) [3].

The primary process that leads to atherosclerosis is represented by endothelial dysfunction, and the organism is trying to counterbalance it through the activation of progenitor endothelial cells to produce vascular repair.

Inflammation plays a key role in PAD and CLTI, but its mediators are not clearly defined [4,5]. The role of inflammatory biomarkers in the atherosclerotic process has been extensively analyzed, both in experimental and clinical studies. Although inflammatory mediators such as IL-6, TNF-α, and C-reactive protein (CRP) were identified as predictors for major cardiovascular events, randomized studies demonstrated opposite results, which justifies a more thorough analysis of these markers in CLTI. 

It is well known that in the early stages of the atherosclerotic process, endothelial dysfunction leads to the production of increased quantities of inflammatory cytokines (IL1β, IL6, CRP, MCP-1—monocyte chemoattractant protein-1), proteases (matrix metalloproteinase—MMP), extracellular vesicles, and adhesion molecules (P-selectin, E-selectin, ICAM—Intracellular Adhesion Molecule, VCAM—Vascular Cell Adhesion Molecule), which contributes to monocyte attraction and is secondary to intima infiltration. However, compared with coronary artery disease, these mechanisms are not widely studied for CLTI [6].

The aim of our review was to summarize the most important markers studied so far in CLTI, keeping in mind that algorithms derived from these markers can lead to prediction models in PAD, CLTI, and major amputations, or can estimate the risk for major cardiovascular events in patients with CLTI.

## 2. Materials and Methods

We performed an electronic medical database search—of Web of Science, MEDLINE, and Embase—in which we included English-language articles from January 1956 to November 2023. We also analyzed the references in every article. The terms we used were: critical limb ischemia, chronic limb-threatening ischemia, lower-limb ischemia, inflammatory markers, C-reactive protein (CRP), D-dimers, fibrinogen, cytokines, interleukin-6 (IL-6), Tumor Necrosis Factor-α (TNF-α), Intracellular Adhesion Molecule-1 (ICAM-1), Vascular Cell Adhesion Molecule-1 (VCAM-1), neutrophile-to-lymphocytes ratio (NLR), Interleukin 8 (IL-8), Pentraxin-3 (PTX-3), neutrophil gelatinase-associated lipocalin (NGAL), calprotectin, E-selectin, P-selectin, neopterin, High-Mobility Group Box-1 protein (HGMB-1), Osteoprotegerin (OPG) and Sortilin (SORT) (Figure 1). 

The search results were imported into EPPI-Reviewer 6 for duplicate removal and study selection. After this step, titles and abstracts were independently reviewed by two investigators (A.I.P and A.L.R.). Full-text articles were then read, and we collected data on the study type, methods, participants, outcomes, and results of every different study. All disagreements were solved by discussion between all the authors. 

This review will focus on the association of the above markers with chronic limb-threatening ischemia in terms of prediction, progression, major cardiovascular events in these patients, and major limb events, but it will also emphasize other specific findings when considered useful to clinical practice.

## 3. Results

When speaking of chronic threatening limb ischemia, we are speaking of the following marker categories: markers of thrombosis, markers of lipid metabolism, markers of tissue remodeling and angiogenesis, inflammatory markers, markers of endothelial activation and dysfunction, and markers of oxidative stress. 

We have attempted to synthesize the most important markers and highlight the most relevant studies related to CLTI.

### 3.1. C-Reactive Protein (CRP)

C-reactive protein (CRP) was identified in 222 studies for PAD in general and in 8 studies for CLTI. CRP is an acute-phase reactant, and its plasma concentration can increase up to 1000-fold in the acute phase of the pathology [7]. The predictive association between CRP and coronary heart disease has been extensively demonstrated. Different studies have demonstrated that CRP contributes to inflammation at the level of atheroma plaque and is actively involved in early atherogenesis [8]. hs-CRP (high-sensitivity CRP) is known as a predictive factor of myocardial infarction [9,10], stroke, and peripheral arterial disease [11,12,13].

Patients with peripheral arterial disease have elevated levels of hs-CRP, and it appears to be predictive of disease progression, as measured by the ankle–brachial index, and the severity of functional impairment [14].

Nardella et al. [15] conducted a prospective non-randomized study of 264 patients in which they identified the relationship between baseline CRP levels and the incidence of major cardiovascular events (MACEs) and major lower-limb events (MALEs) in diabetic patients with CLTI. They identified elevated CRP levels with poor post-revascularization outcomes in patients with CLTI and diabetes. Moreover, 15.9% of patients developed MACEs within the first year after revascularization and all had elevated serum CRP values. Likewise, 30.7% of patients developed MACEs 1 year after revascularization and, again, serum CRP levels were significantly elevated. The authors concluded from multivariate logistic regressions that in patients with critical ischemia, elevated CRP values are an independent determinant of MALEs and MACEs after revascularization. 

Husakova J. et al. [16] studied risk factors for major amputations in patients with diabetic ulcers and CLTI treated with autologous cell therapy. Their study included 113 patients divided into two groups—those with and those without major amputation. The group with major amputation had significantly increased CRP levels compared to the group without major amputation (22.7 vs. 10.7 mg/L).

Another study led by Fujii M. retrospectively studied 30 patients with CLTI and forefoot osteomyelitis and recommends, as the first therapeutic intention, debridement and infection clearance if a CRP level greater than 40 mg/L is identified [17].

Biscetti et al. [18] studied the association between the levels of the most important inflammatory cytokines and the presence of diabetes mellitus in a group of 299 patients, as well as the results of these markers after endovascular procedures in patients with diabetes mellitus and CLTI. They identified increased levels of CRP values in association with a negative postoperative outcome in patients, with an association between CLTI and diabetes.

In a prospective study of 71 patients with Fontaine stage II-IV PAD, Tschopl et al. [19] studied the role and values of various known hemostatic factors that may be associated with an increased risk of thrombosis and identified that patients with restenosis of atherosclerotic lesions had higher CRP values at 3–6 months after PTA (25.4 +/− 46.7 versus 7.9 +/− 6.9 mg/L, *p* < 0.05).

Also, another observational study of 116 patients, of which 46% had CLTI, showed increased serum values in these patients compared to patients without CLTI (37.53 ± 46.61 mg/L vs. 9.18 ± 26.12 mg/L, *p* = 0.000) [20].

### 3.2. D-Dimers

D-dimers are a degradation product of cross-linked fibrin and are a marker of coagulation system activation. They are a marker of fibrinolysis and are often associated with thrombosis processes [21,22]. Vidula et al. [23] reported a relation between D-dimer levels and mortality in patients with PAD, as well as higher levels 1–2 years before death. Elevated D-dimer levels have been associated with a risk of subsequent venous or arterial thrombotic events, especially in patients with pre-existing vascular disease [22,24].

Tschopl et al. also studied D-dimer output in Fontaine stage II-IV patients and found significantly increased D-dimer values at 1 h, 24, and 48 h, respectively, after PTA in a group of 71 patients [19].

In a systematic review including 10 studies and 2420 patients, other authors demonstrated that increased D-dimer levels are associated with a two-fold increased risk of developing major cardiovascular events in patients with PAD [24].

### 3.3. Fibrinogen 

Fibrinogen is a glycoprotein with an extremely important role in several physiological and biochemical processes. It is expressed primarily in hepatocytes, and its synthesis is regulated by acute-phase proteins. Thus, fibrinogen is an acute-phase reactant because its synthesis is increased during inflammation, in the acute phase, its plasma concentration can exceed 7 g/L [25]. Fibrinogen synthesis is stimulated by cytokines in activated megakaryocytes and plays a key role as an acute-phase reactant. Thus, fibrinogen is considered a key molecule in the inflammation and coagulation cascade and is involved in the pathogenesis of the atherosclerotic process and the development of atherothrombotic complications [26].

In their study, Tschopl et al. [19] demonstrated significantly increased fibrinogen values 48 h after PTA. Another phenomenon demonstrated by them is that patients who experienced post-revascularization restenosis showed increased values of this marker, both initially and 3–6 months after PTA.

In an observational prospective study on 116 patients, Ferreira et al. [20] demonstrated that patients with CLTI had increased serum fibrinogen levels compared to patients without CLTI (466.18 ± 208.07 mg/dL vs. 317.37 ± 79.42 mg/dL, *p* = 0.000).

In a study of 108 CLTI patients led by Pedrinelli et al. [27], who studied the association between mortality rate and plasma fibrinogen levels in CLTI patients who died during the first 6 months of follow-up, extremely elevated plasma fibrinogen values were identified. Also, in a Cox-multivariate regression analysis, only plasma fibrinogen had an independent predictive value for mortality.

### 3.4. Interleukin-6 (IL-6) 

Interleukin-6 (IL-6) was discovered in 1986 and is a member of the pro-inflammatory cytokine family; it induces the expression of proteins responsible for the acute phase of inflammation and plays an important role in cell differentiation and the proliferation of non-immune cells [28]. IL-6 also plays an important role in the acquired immune response by stimulating antibody production and the development of effector T-cells [28].

An important element to note about IL-6 that is directly related to critical ischemia is that an immediate expression of IL-6 is generated in response to environmental stressors such as infection or tissue injury. Withdrawal of the source of stress on the host will lead to a decrease in the activation of the IL-6-mediated negative regulatory system cascade, with degradation of IL-6 mRNA via regenerase-1 leading to a termination of IL-6 production. A variety of circumstances related to the vascular wall, such as oxidative stress and vascular injury, can lead to increased IL-6 secretion [29]. There is also a strong association between circulating IL-6 levels and the risk of major cardiovascular events [30,31,32].

Nardella et al. demonstrated in a prospective study of 264 patients that elevated IL-6 levels are associated with poor outcomes in diabetic patients with critical ischemia after revascularization [15].

Gremmels et al. [33] investigated several markers with a role in the occurrence of major cardiovascular events or major amputations. IL-6 was found to be significantly elevated in patients who reached one of the above points, leading to the conclusion that inflammatory markers play a role in predicting the risk of major cardiovascular events in patients with critical ischemia.

### 3.5. Tumor Necrosis Factor-α (TNF-α)

Tumor Necrosis Factor-α (TNF-α) is a multifunctional cytokine that has been implicated as a mediator of various physiological and pathophysiological events, including cell survival, growth, differentiation, apoptosis, inflammation, and angiogenesis [34].

Two receptors with high affinity for TNF-α are TNF-α-Receptor-1 (TNF-α-R-1, known as p55) and TNF-α-Receptor -2 (TNF-α-R-2, known as p75), with roles in regulating processes in critical ischemia [34].

Caicedo et al. [35] studied TNF-α in 35 patients with CLTI and found increased levels (≥8.1) in 68.6% of patients and concluded that this marker is very important in the progression of ischemia, especially in diabetic patients.

Also, Biscetti et al. [18] showed in 299 patients with BTK disease that TNF-α is associated with worse vascular outcomes in patients with CLTI and diabetes mellitus.

### 3.6. Intracellular Adhesion Molecule—1 (ICAM-1) and Vascular Cell Adhesion Molecule—1 (VCAM-1)

Soluble Intracellular Adhesion Molecule-1 (ICAM-1), also known as CD54, encodes a cell surface glycoprotein that mediates leukocyte stationary adhesion and transmigration and can be regulated during tissue reperfusion. A study led by Foussard et al. [36] demonstrated that impaired limb reperfusion was associated with increased endothelial activation of ICAM-1 expression and impaired perfusion of capillaries that are smaller in diameter than a leukocyte when it interacts with leukocytes. 

VCAM-1 is expressed in endothelial cells in response to cytokines, mediates the adhesion of leukocytes, including lymphocytes and monocytes, and is associated with vascular inflammation. VCAM-1 also plays a key role in leukocyte–endothelial cell signal translation and in the development of atherosclerosis [37]. Regarding the relationship between VCAM-1 and PAD, patients with PAD are more likely to have increased circulating levels of soluble adhesion molecules [38].

When searching the databases, we did not specifically find the association between the two molecules and CLTI specifically. However, we have identified various studies related to PAD in general and the association with the 2 molecules. Lee et al. [39] identified elevated serum levels and tissue expression of VCAM-1 as increased in patients with PAD and advanced chronic kidney disease.

Another study by Edlinger et al. [40] on 126 patients, of whom 51 with PAD manifestations and 75 without a history of PAD, demonstrated increased VCAM-1 values in patients with PAD, 1352 pg/mL vs. 953 pg/mL) and concluded that VCAM-1 is a marker that accurately identifies the severity of the systemic atherosclerotic process.

### 3.7. Neutrophile-to-Lymphocytes Ratio (NLR)

Over the last 10 years, this ratio seems to have become an extremely important marker. It is calculated as the simple ratio of neutrophils to lymphocytes measured in peripheral blood and is a marker that shows two sides of the immune system: the innate side due to neutrophils and the acquired side given by lymphocytes [41].

Neutrophils are responsible for the first line of immune response by chemotaxis, phagocytosis, and the release of oxidative reactive species, granular proteins, and cytokines [42]. NLR has been associated with high mortality in patients with critical ischemia [42].

We found 26 articles and selected 11 that addressed chronic limb-threatening ischemia.

A retrospective study of 268 patients who were not scheduled to receive either surgical or endovascular treatment but only a conservative treatment showed that patients with a ratio greater than 4.63 did not respond positively to drug treatment of the pathology [43].

Gary T. et al. [44] demonstrated in another retrospective study of 2121 patients with PAD at various stages that the neutrophil/lymphocyte ratio was significantly increased in patients with CLTI (usually around 3.95).

Another retrospective study of 195 patients in which NLR was associated with critical ischemia showed that at 1-year follow-up, patients with NLR over 8 had higher in-hospital mortality. In terms of major amputation events, an NLR ≥ 6 was associated with a higher frequency (adjusted HR: 2.804, 95% CI: 1.292–6.088, *p* = 0.009) [45].

Spark et al. [46], in a prospective study of 149 patients with an average follow-up period of 8.7 months, identified that increased NLR was identified in a group of patients with critical ischemia at risk of developing major amputation-type events.

### 3.8. Interleukin-8 (IL-8)

IL-8, also known as neutrophil chemotactic factor, was first discovered in 1987; its primary role is to induce chemotaxis in neutrophils to the damaged tissue [47]. The main cellular sources of IL-8 are typically monocytes and macrophages, with IL-8 playing a primary role in the recruitment of monocytes and neutrophils—promoters of the acute inflammatory response. A functional feature of IL-8 is that it remains for a very long time at the site of acute inflammation, sometimes even weeks; this contrasts with other inflammatory cytokines that disappear within hours. IL-8 is very sensitive to anti-oxidants, which greatly reduce IL-8 gene expression, and has an important role in vascular diseases, where ischemia-induced oxidative stress is both a marker of disease and a potential target for therapy [48,49].

In a study comparing critical ischemic patients with PTA vs. diagnostic angiography (25 vs. 20 patients), baseline IL-8 levels pre- and post-procedurally demonstrated that IL-8 values were not significantly different between the two groups, and at 1 year, 45% of the PTAs failed, but there was no significant correlation with the inflammatory markers [50].

Another observational, prospective study of 119 patients, including 45 with CLTI, which analyzed histopathological samples from the sartorius muscle of 26 patients, demonstrated significant associations with increased IL-8 levels in these patients [51].

### 3.9. Pentraxin-3 (PTX3)

PTX3 is an inflammatory phase glycoprotein from the same family as CRP. It appears to be localized in atheroma plaque [52]. The synthesis of PTX3 is stimulated in endothelial cells, macrophages, myeloid cells, and dendritic cells by cytokines, endotoxins such as bacterial products, IL-1, and TNF. PTX3 is localized in the atheroma plaque and plays an important role in modulating the procoagulant activity of endothelial cells [53].

A literature review demonstrated the association of the presence of elevated PTX-3 values with the severity of peripheral arterial disease but did not specifically discuss critical ischemia [5].

A retrospective study of 12 patients demonstrated the association of PAD severity with PTX-3 values, and the authors extend the idea that this marker could be used in relation to the Rutherford classification to determine PAD severity [54].

### 3.10. Calprotectin and Neutrophil Gelatinase-Associated Lipocalin (NGAL)

Calprotectin is a complex protein that, in the presence of calcium, is capable of binding transition metals such as iron, zinc, and manganese, thus offering complex antimicrobial activity [55]. An association has been identified between increased circulating levels of calprotectin and the risk of developing cardiovascular events or amputation in patients with symptomatic BAP [5]. Saenz-Pipaon et al. [5] studied 331 Fontaine II-IV patients (serum levels) and 413 Fontaine 0–1 patients (plasma levels). They analyzed human femoral plaque and identified that calprotectin levels were 64% lower in plasma than in serum. In PAD patients, the risk of major limb events and major cardiovascular events was associated with 1.8-fold higher serum values for calprotectin [55].

NGAL is a polypeptide released by damaged nephron tubular cells, which is also expressed in atheromatous plaque and is implicated in the development of cardiovascular disease. Various studies have shown that increased levels of NGAL are correlated with the severity of coronary heart disease [56] and are also seen in obese and diabetic patients and correlate with increased CRP levels [57].

### 3.11. E-Selectin (CD-62)

The leukocyte–endothelial cell adhesion molecule represents an adhesion molecule expressed only in cytokine-activated endothelial cells. Like other selectins, it plays an important role in inflammation, being vital for the recruitment of progenitor stem cells required for the neovascularization of ischemic tissue. Animal studies have shown that E-selectin is vital for neovascularization and tissue repair [58].

A major focus is now on preclinical studies using mesenchymal stem cells enhanced with soluble E-selectin to enhance ischemic tissue repair and regeneration in animal models. In 2022, Quiroz et al. [59] created a viral vector to overexpress E-selectin on mesenchymal cells in order to increase their therapeutic profile; they concluded that the new “supercharged” cells grow faster and induce functional recovery, angiogenesis, and tissue repair in CLTI murine models.

A study of 157 patients with PAD and 206 in the control group that analyzed genetic polymorphism demonstrated that E-selectin was independently associated with PAD, and synergistic effects were identified, resulting in a genetic profile that significantly influences the increased risk of CLTI [60].

Another prospective study of 175 patients with PAD (one group with intermittent claudication and one group with CLTI) who underwent endovascular intervention procedures demonstrated that patients with E-selectin values above 44.9 mg/dl had a 1.9-fold adjusted risk of restenosis (95% CI 1.09 to 3.30) [61].

### 3.12. Neopterin

Neopterin is a pyrazino–pyrimidine compound that may play an important role in the pathogenesis and progression of coronary artery disease and peripheral arterial disease by regulating bioavailability and the proliferation and differentiation of hematopoietic stem cells [62].

Some studies have demonstrated the association of neopterin with ABI and increased mortality in patients with vascular calcifications undergoing angiography [63].

Despite all the above markers, neopterin has been studied more extensively in relation to CLTI. Neopterin levels were also found to predict vascular adverse events in patients with CLTI, making this molecule a very useful marker of atherosclerotic disease [64].

Signorelli et al. [65] investigated neopterin levels in three groups of patients (control group, symptomatic PAD, and asymptomatic PAD) and demonstrated that mean plasma levels were higher in symptomatic (9.4 ± 4.6 nmol/L) and asymptomatic (7.4 ± 4.0 nmol/L) patients compared to control patients (5.3 ± 3.2 nmol/L).

Another observational study of 259 patients with critical ischemia studied neopterin levels in diabetic vs. non-diabetic patients and found increased levels in diabetic patients (31 vs. 21 mmol/L, *p* < 0.01) [66].

### 3.13. High-Mobility Group Box-1 (HMGB-1)

This is a nuclear protein that regulates gene expression and induces an inflammatory response during vascular injury, but it also plays a role in activating the pro-inflammatory response after it is released passively by necrotic cells or actively secreted by mock immune cells such as endothelial cells [67]. Moreover, the hs-CRP-induced release of HMGB-1 induces, amplifies, and extends the inflammatory process surrounding the atherosclerotic injury. Further, HMBG-1 causes the secretion of inflammatory molecules such as IL-6 and TNF-α by neutrophils and macrophages. In diabetic patients, this protein promotes chronic inflammation and neovascularization, is directly correlated with HbA1c levels, and is present in the retina of diabetic retinopathy patients [68].

Nardella et al. [15] studied the relationship between baseline biomarker levels and the incidence of MACEs (major adverse cardiac events) and MALEs (major adverse limb events) in diabetic patients with CLTI and identified that increased levels correlate with poor post-revascularization outcomes.

A prospective observational study of 201 CLTI patients who required revascularization benefited from HGMB-1 determination, and the aim of the study was to assess the association of HGMB-1 with MACEs or MALEs after revascularization. Patients who developed limb-related events or major cardiovascular events after revascularization had higher HGMB-1 serum levels: 7.5 ng/mL vs. 4.9 ng/mL (*p* < 0.01) for cardiovascular events and 7.2 ng/mL vs. 4.8 ng/mL (*p* < 0.01) for limb events [69].

### 3.14. Osteoprotegerin (OPG)

This is a member of the TNF family of receptors involved in bone turnover, and recent studies have shown that it is an important molecule in the atherosclerotic process and contributes to the calcification of small vessels in diabetic patients. OPG expression is induced by HMBG-1 in osteoblasts and attaches to osteoclasts and osteoblasts during endochondral ossification [70].

Current studies have generated quite controversial results. At this point, its role and relationship with inflammatory markers in diabetic patients with peripheral artery disease is quite unclear, and further investigations are needed [71].

All the studies demonstrated that increased levels of OPG correlate with poor outcomes in diabetic patients with CLTI after revascularization [15,72].

Giovannini et al. [73], in a retrospective observational study that analyzed OPG levels in 1393 diabetic patients with and without PAD, demonstrated that OPG levels were significantly higher in PAD vs. non-PAD patients (6.54 ± 7.76 pmol/L vs. 2.98 ± 2.01 pmol/L) and that OPG levels also correlated positively with the clinical severity of PAD.

### 3.15. Sortilin

Sortilin is a protein that is mainly expressed in hepatocytes; it takes part in the sorting of intracellular proteins between the Golgi network and endocytosomes. The sortilin membrane acts as a receptor for circulating LDL and promotes its integration into hepatocytes via a receptor-dependent mechanism. Following the discovery of the link with the atherosclerotic process, key roles in cytokine release or expression in platelets have also been discovered. It is also involved in LDL uptake by macrophages and the progression of inflammatory mechanisms during the process of atheroma plaque formation as well as its progression [74,75].

Nardella et al. [15] also demonstrated in 264 patients that increased levels of sortilin correlate with poor outcomes in diabetic patients with CLTI after revascularization.

All our findings are summarized in Table 1. 

Also, in Table 2, we detail the mechanisms and effects of these markers, as well as the specific method for laboratory determination.

## 4. Discussions

The atherosclerotic process begins with endothelial dysfunction based on the concurrent action of multiple risk factors (hypertension, diabetes, obesity, high LDL, chronic kidney disease, or other inflammatory diseases) [5]. Following damage to the vascular endothelium, a series of pro-inflammatory cytokines (IL6, CRP), extracellular vesicles, a series of proteases, and adhesion molecules (E-selectin, ICAM, VCAM) are released, which recruit monocytes and infiltrate the vascular lining [18]. Another phenomenon occurring currently is the activation of environmental factors that contribute to atheroma plaque progression through macrophage polarization. An important role is also played by extracellular vesicles of thrombocytic origin, which play an extremely important role in thrombosis phenomena in the atheroma plaque in the context of exposure to procoagulant factors. Thrombotic phenomena in turn lead to increased levels of biomarkers such as CRP, ICAM, VCAM, interleukin family, or vascular smooth muscle cells (VSMCs) [79].

Peripheral arterial disease, and in particular its end stage, CLTI, is associated with significant morbidity, mortality, and a serious impairment of quality of life. CLTI is often accompanied by atherosclerotic processes in all vascular beds, and this association presents a significant risk of ischemic events and death in these patients, much more than the single impairment in other vascular beds [5].

Currently, PAD risk assessment is performed only on clinical classifications and an ABI basis and is not highly specific, especially in patients with concomitant diabetes mellitus (where we know all too well of the presence of false-positive ABI values due to massive calcifications of infragenicular vessels). 

In this context, PAD is an often-underestimated pathology, especially in patients with diabetes, thus leading to the need for early diagnostic and prognostic markers.

Not all markers presented have sufficient studies to be considered diagnostic or prognostic markers in CLTI. Of those we present, CRP, fibrinogen, D-dimer, IL-6, IL-8, and NLR have multiple studies that demonstrate their value. We consider that the others need further studies to prove their involvement in the atherosclerosis process and the possibility of their inclusion as diagnostic or prognostic markers for major cardiac events or events related to the performance of revascularization procedures. 

Besides the markers studied and presented in this review, it would be more important to pay attention to extracellular vesicles and micro-RNAs that would show an effect in PAD and CLTI.

There is not enough information in the literature related to biomarkers present in CLTI, but the discovery of these links would clearly contribute to the discovery of new diagnostic and therapeutic methods for these patients.

There are several studies focusing on the identification of some types of microRNAs in CLTI, such as miR-1827, miR-323b-5p, miR-4739, and miR-27b, which are significantly elevated in the serum of these patients. It is also worth studying the interaction between other circulating elements and micro-RNAs in different stages of atherosclerosis [80].

Another important phenomenon that needs to be discussed when speaking about atherosclerosis and CLTI relates to the extremely important role that the innate and adaptive immune system appears to play in this process. The attraction of cytokines and the expression of adhesion molecules seem to trigger a cascade response of immune cells, which leads to the nucleation of a lesion in the vascular wall that may subsequently develop into an atherosclerotic plaque. In addition, dendritic cells, macrophages, and T cells (all belonging to the immune system) appear to play a role in the development of the atheroma plaque. Furthermore, along the way, these cells undergo apoptosis and must be eliminated, leading to secondary necrosis and the formation of a necrotic core in the plaque. Programmed cell death at this level can also lead to the progression of atherosclerosis. Plaque development is favored by the tipping of the scales toward autoimmune cell recruitment and continued macrophage proliferation [81].

Atherosclerotic disease is considered to have a multifactorial etiology that includes both genetic determinism and the action of environmental factors [82].

There is current evidence of the importance of genetic factors through various population studies, with an estimated heritability of PAD of approximately 20–58% [83]. There are currently a number of genes that are involved in atherosclerosis: genes that are involved in lipid metabolism (*APOB* on chromosome 2p24.1), genes involved in inflammatory mechanisms in the vascular wall (IL6 gene located on 7p15.3), genes involved in blood pressure regulation (*ACE* genes), genes involved in the function of Vascular Smooth Muscle Cells, genes involved in the maintenance of vascular homeostasis, or genes that contribute to the process of atherothrombosis. The discovery of the genes involved in the pathophysiological mechanisms of BAP is a step toward finding an early diagnosis as well as a complete and effective treatment of this pathology [84].

## 5. Conclusions

This review gathered data on inflammatory markers with a role in peripheral arterial disease in general and critical ischemia in particular, noting that the search did not identify exact associations between inflammatory markers and what current guidelines define as critical ischemia.

Therefore, we believe that further studies are needed to identify the targeted feasibility of using inflammatory markers in the diagnosis, prognosis, and treatment of critical ischemia in practice.

Thus, we need studies to identify different risk classes according to the value of the markers and to identify the disease prognosis according to the evolution of these markers. Also, we need to identify these markers in the clinical staging of the disease in the post-revascularization phase, both in terms of events related to the revascularized limb and in terms of major cardiac events.

## Figures and Tables

**Figure 1 biomedicines-12-00798-f001:**
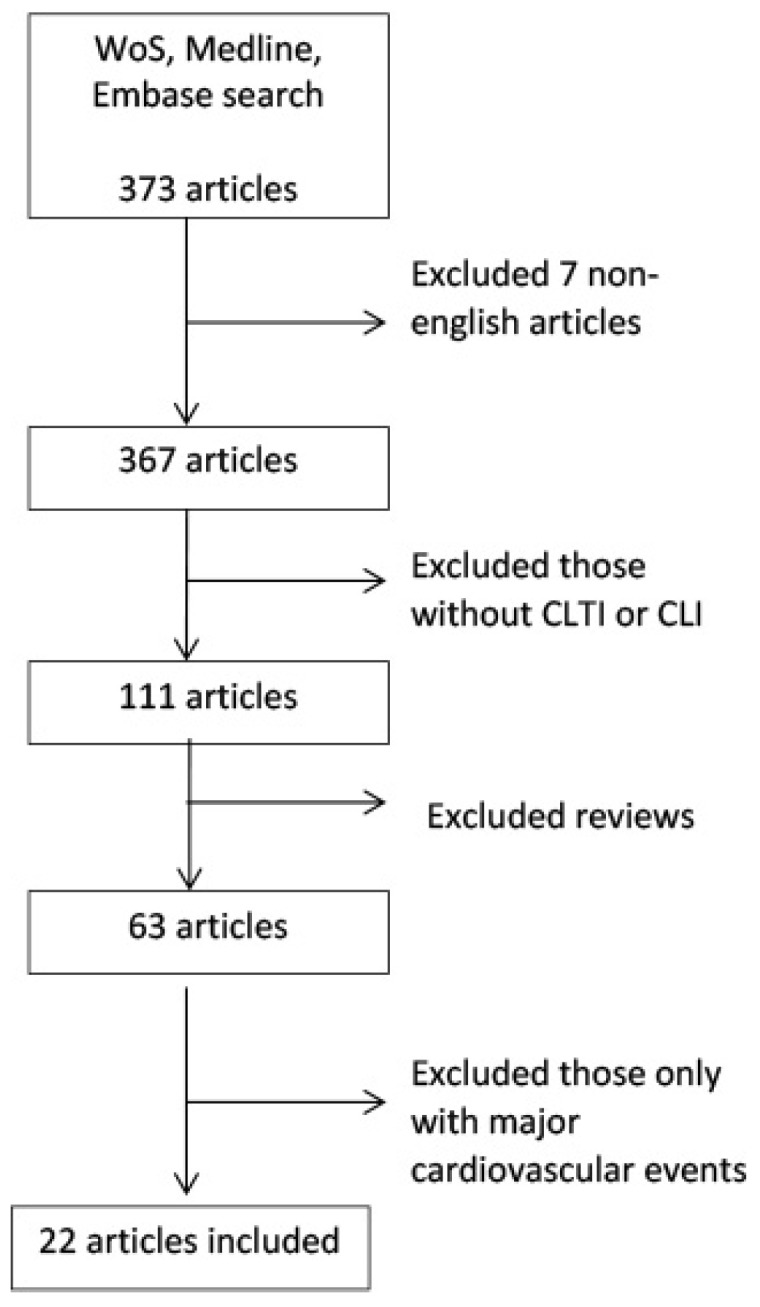
PRISMA flow diagram.

**Table 1 biomedicines-12-00798-t001:** Summary of biological markers found in CLTI.

Study	Author	Marker(s)	Population	Findings
Development of a biomarker panel for assessing cardiovascular risk in diabetic patients with chronic limb-threatening ischemia (CLTI): a prospective study	Nardella et al., 2023 [15]	CRP, IL-6, HMGB-1, OPG, Sortilin	264 CLTI patients	Increased levels correlate with poor outcomes in diabetic patients with CLTI after revascularization
Factors Influencing the Risk of Major Amputation in Patients with Diabetic Foot Ulcers Treated by Autologous Cell Therapy	Husakova et al., 2022 [16]	CRP	113 patients with diabetic foot ulcers	Increased levels of CRP in the group that suffered major amputation than in the group that did not
Inflammatory Cytokines Associated with Failure of Lower-Extremity Endovascular Revascularization (LER): A Prospective Study of a Population With Diabetes. Diabetes Care	Biscetti et al., 2019 [18]	CRP, TNF-α	299 patients	Elevated CRP levels in patients with worse vascular outcomes in patients with diabetes TNF-α is associated with worse vascular outcomes especially in diabetic patients
Role of hemostatic risk factors for restenosis in peripheral arterial occlusive disease after transluminal angioplasty	Tschopl et al., 1997 [19]	CRP, D-dimers, Fibrinogen	71 patients	Restenotic patients had higher CRP levels at 3 and 6 months after PTA Increased levels of D-dimers at 1, 24 and 48 h after PTA Fibrinogen levels higher 48 h after PTA. Restenotic patients had higher plasma fibrinogen levels at follow-up. Patients with III/IV stages had significantly higher fibrinogen values
Inflammation and Loss of Skeletal Muscle Mass in Chronic Limb Threatening Ischemia	Ferreira et al., 2023 [20]	CRP, Fibrinogen	116 patients, 46% with CLTI	Increased CRP serum levels vs. non-CLTI patients CLTI patients had increased fibrinogen levels than non-CLTI patients
Fibrinogen and mortality in chronic critical limb ischemia	Pedrinelli et al., 1999 [27]	Fibrinogen	108 patients, 78 with CLTI	48 deceased patients from CLTI group, with increased fibrinogen values in the first 6 months on follow-up and in a Cox-multivariate analysis only plasma fibrinogen had an independent predictive value
A Pro-Inflammatory Biomarker-Profile Predicts Amputation-Free Survival in Patients with Severe Limb Ischemia	Gremmels et al., 2019 [33]	IL-6	108 patients	Il-6 values were higher in patients that reached a major endpoint (death and/or major amputation)
The Inflammatory Pattern of Chronic Limb-Threatening Ischemia in Muscles: The TNF-α Hypothesis	Caicedo et al., 2022 [35]	TNF-α	35 patients	Increased levels in 68.6% of patients and concluded that the marker is important in the progression of ischemia, especially in diabetic patients
Expression of Vascular Cell Adhesion Molecule-1 in Peripheral Artery Disease is Enriched in Patients with Advanced Kidney Disease	Lee et al., 2021 [39]	VCAM-1	51 patients	Elevated levels
Critical limb ischemia patients clinically improving with medical treatment have lower neutrophil-to-lymphocyte and platelet-to-lymphocyte ratios	Erdoğan et al., 2021 [43]	NLR	268 patients	Patients with NLR higher that 4.64 did not respond to medical treatment
Neutrophil-to-lymphocyte ratio and its association with critical limb ischemia in PAOD patients	Gary et al., 2013 [44]	NLR	2121 patients	Increased values in patients with CLTI
Neutrophil-to-lymphocyte ratio associated with an increased risk of mortality in patients with critical limb ischemia	Su et al., 2021 [47]	NLR	195 patients	Increase levels associated with major cardiovascular or major limb events
An elevated neutrophil-lymphocyte ratio independently predicts mortality in chronic critical limb ischemia	Spark et al., 2010 [46]	NLR	149 patients	Elevated levels in patients with higher risk at developing major adverse limb events
Inflammation Is a Histological Characteristic of Skeletal Muscle in Chronic Limb Threatening Ischemia	Ferreira et al., 2024 [51]	IL-8	119 patients	Higher serum levels
Relationship of Inflammatory Biomarkers with Severity of Peripheral Arterial Disease	Igari et al., 2016 [54]	PTX3	89 patients	PTX3 highly correlated with the severity of the disease
Lipocalin-2 and Calprotectin Potential Prognosis Biomarkers in Peripheral Arterial Disease	Saenz-Pipaon et al., 2022 [55]	Calprotectin and lipocalin-2	331 patients	High serum levels associated with major limb and major cardiovascular events in CLTI patients
Pro-inflammatory genetic profiles in subjects with peripheral arterial occlusive disease and critical limb ischemia	Flex A. et al., 2007 [60]	E-selectin	157 patients	E-selectin polymorphism associated with CLTI
E-Selectin and restenosis after femoropopliteal angioplasty: prognostic impact of the Ser128Arg genotype and plasma levels	Mlekusch et al. 2004 [61]	E-selectin	175 patients	High values associated with restenosis risk in CLTI patients
Inflammatory mediators are associated with 1-year mortality in critical limb ischemia	Barani et al., 2005 [76]	Neopterin	256 patients	Associated with 1-year mortality in CLTI patients
Are there differences of inflammatory bio-markers between diabetic and non-diabetic patients with critical limb ischemia?	Bertz et al., 2006 [66]	Neopterin	259 patients	High values in diabetic patients with CLTI
Serum high mobility group box-1 levels associated with cardiovascular events after lower extremity revascularization: a prospective study of a diabetic population	Rando et al., 2022 [69]	HGMB-1	201 patients	High serum levels in CLTI patients that developed limb related events or major cardiovascular events
Osteoprotegerin concentration is associated with the presence and severity of peripheral arterial disease in type 2 diabetes mellitus	Demková K. et al., 2018 [72]	OPG	165 patients	High levels associated with the severity of CLTI

**Table 2 biomedicines-12-00798-t002:** Summary of actions and effects of the studied markers.

Biomarker	Marker Type	Action and Effects	Methods for Identification
CRP	Inflammatory	- Inhibits the nitric oxide and Increases endothelin-1 production contributing to endothelial dysfunction (it impairs endothelial-dependent vascular relaxation) - Contributes also to thrombus formation - Induces leukocyte recruitment into atherosclerotic tissue	Immunoturbidimetric determination
D-dimers	Thrombotic	- Increases fibrin turnover	Coagulometry
Fibrinogen	Thrombotic	- Pro-inflammatory	Coagulometry
Interleukin-6	Inflammatory	- Pro-inflammatory cytokine, released by leukocytes and stromal cells	Chemiluminescence
Tumor Necrosis Factor	Inflammatory	- Inflammatory factor probably through upregulating the transcytosis of LDL across endothelial cells and retaining the LDL-particles in vessel walls [77]	ELISA Technique
ICAM-1 and VCAM-1	Inflammatory	- Endothelial adhesion - Inflammatory molecules	Immunoassay determination Flow-cytometric method ELISA Technique
NLR	inflammatory	- Neutrophils may make atherosclerotic plaque more vulnerable and secrete inflammatory mediators	Flow-cytometric method, spectrophotometry
Interleukin-8	Inflammatory	- It is involved by recruiting inflammatory cells into the atherosclerotic plaques	Chemiluminescence ELISA Technique
Pentraxin-3	Inflammatory	- Similar to CRP, being a cognate molecule of CRP - Regulatory role in inflammation and extracellular matrix organization and remodeling [78]	Immunoturbidimetric determination
Calprotectin and NGAL	Inflammatory	- Pro-inflammatory	ELISA Technique
E-selectin	Inflammatory	- Adhesion molecule mediating leukocyte extravasation	Multiplex immunoassay determination Flow-cytometric method ELISA Technique
Neopterin	Inflammatory	- Is a metabolite of guanosine triphosphate with proinflammatory action	Immunoenzymatic assay
HMGB-1	Inflammatory	- Mediates inflammation	ELISA Technique
Osteoprotegerin (OPG)	Inflammatory	- Acts in endothelial dysfunction as a proinflammatory and pro-calcification marker	ELISA Technique
Sortilin	Inflammatory	- Arterial wall inflammation and calcification	Flow-cytometric method

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
