# Peer review of "Narrative Review of Biological Markers in Chronic Limb-Threatening Ischemia"

_biomedicines, 2024, doi:10.3390/biomedicines12040798_

Round 1

Reviewer 1 Report

Comments and Suggestions for Authors

The presented for peer-review the review paper: 'Narrative Review of Biological Markers in Chronic Limb Threatening Ischemia' is interesting and well-written. The Authors have presented many important biomarkers, cytokines and molecules identified through the electronic medical database search – Web of Science, MEDLINE and Embase focused on the PAD outcomes. In total, 22 papers were chosen for this narrative review.

I find this paper important and well-presented. The presented biomarkers, including hs-CRP, fibrinogen, D-dimers, TNF-alpha, IL-6, -8, Pentraxin-3, NGAL, calprotectin, E-selectin, P-selectin, neopterin, HGMB, Osteoprotegerin and Sortilin are all important.

1) However, the electronic search showed that some data on the individual biomarker have limited level of evidence. And, this is my major concern - the Authors should address this issue in at least short Discussion parahraph which is now missing, leaving the Readers without a final feedback. Please, specify which biomarkers have high level of evidence for its credibility, which data are limited.

2) I believe thatit is important to explain why the Authors focused on PAD, not on the other arterial territories like coronary, carotid or renal. I think that the following review is suitable to address the diversities in atherosclerosis drivers between arterial territories: https://doi.org/10.3390/jcm13051471

In particular, in PAD, huge roles of smoking toxicity and diabetes are the most important drivers of natural course of atherothrombosis in limb ischemia, in contrast to hiperlipidemia and hypertension in coronary and carotid arterial beds.

3) There is only one Table: Table 1. Summary of biological markers found in CLTI. I am convinced that an illustration about PAD and biomarkers having role in PAD and CLI would enhance this review, or adding Table 2 on the pathway mechanisms that exert the identified biomarkers on the endothelial cells, vascular smooth muscle cells, or platelets aggregation and thrombosis. Or both

4) Finally, please give some clues about future directions and what else could be done to improve diagnostic / prognostic evaluation of the biomarkers. Are there any additional biomarkers, like microRNAs (miR-134, miR-27) that would be worth of testing or they have already evidence on their important value in PAD.

Minor comments:

Please correct Interleukina-6 (IL-6) - to English Interleukin-6 and Interleukin-8

Comments on the Quality of English Language

Minor comments:

Please correct Interleukina-6 (IL-6) - to English Interleukin

Author Response

Thank you very much for your review. Please find attached our responses.

Reviewer 2 Report

Comments and Suggestions for Authors

The paper Biomedicines-2933332 » Narrative Review of Biological Markers in Chronic Limb Threatening Ischemia« is a review of several arbitrarily chosen markers (C reactive protein, D-dimer, fibrinogen, IL-6, IL-8, TNF-alpha, ICAM-1, VCAM-1, neutrophil to lymphocyte ratio, pentraxin-3, neutrophil gelatinase-associated lipocalin, calprotectin, E-selectin, P-selectin, neopterin, high-mobility group  box-1 protein, osteoprotegerin and sortilin) in the context of  chronic limb threatening ischemia relating to the patients’ prognosis.

The paper interesting, but often severely lacks precision.

I have the following specific concerns:

Is this a narrative review or a systematic review? According to the Abstract (line23), it is a systematic review. Please, reconcile the title and the abstract.

 Materials and Methods, line 81. Please, state which Boolean operators (and, or, not) were used in the search.

Materials and Methods, line 83. Why was the term »critical limb threatening ischemia« used as a search term and not chronic limb-threatening ischemia?

Materials and Methods, lines 83-86. Why are some search terms listed in full with the abbreviation in brackets, and some terms as an abbreviation with the full name in brackets?

 Results, lines 102-106. The authors announce the following categories of markers: markers of thrombosis, markers of lipid metabolism, markers of tissue remodeling and angiogenesis, inflammatory markers, markers of endothelial activation and dysfunction and markers of oxidative stress. However, no further attempt is made to classify the described markers into categories. It would be helpful to classify each of the discussed markers in these terms.

Results. CRP, lines 111 -112. reference No. 7 is misquoted. As clinicians know, and as reference No. 7 (N Engl J Med 1999; 340: 448–454) correctly states, CRP can increase up to 1000-fold.

 Results. Fibrinogen, lines 167-170. “Fibrinogen mediates the modulation of coagulation and fibrinolysis by binding to thrombin, conferring anti-thrombin activity. Fibrinogen also contributes to various pathological events, including thrombosis, due to a decrease in its plasma concentration, change in structural properties or polymorphism effect on clot thickness «. These are very strange introductory sentences for a molecule that is the substrate of thrombin in the forming of fibrin, and is an acute phase reactant.  Please, introduce fibrinogen more accurately.

Results.  Interleukin-6, lines 189-191. The introduction of Il-6 is awkward. Is pleiotropic activity really the main feature of IL-6? Why not something like this: Interleukin-6 (IL-6) is a member of the pro-inflammatory cytokine family, induces the expression of a variety of proteins responsible for acute inflammation, and plays an important role in the proliferation and differentiation of cells in humans. (Oncology 2020; 98:131-7).

Results. Interleukin-8, lines 274-275. The first sentence is not a very good description of Il-8. Why  not something like this: Interleukin-8  is a chemoattractant cytokine produced by a variety of tissue and blood cells, that attracts and activates neutrophils in inflammatory regions (J Periodontol 199; 64(5 Suppl):456-60.)

Results. Calprotectin, lines 305-30.  The first sentence needs clarification – Where does the 1.8-fold increase occur, what side effects does it cause, and please, provide the appropriate reference. Calprotectin also needs an introduction, e.g: Calprotectin, a protein complex secreted by neutrophils, has a high affinity for calcium, zinc, iron, and manganese which results in its antimicrobial activity.

 Results. Lines 312-312. the following sentence does not make sense: »In PAD patients the risk of major limb events and major cardiovascular events were associated with high serum values 5.8 and 1.8 fold.« Please, correct and provide the reference.

 Conclusions, lines 415-416. I cannot agree that the described markers are “specific to peripheral arterial disease in general, and critical ischemia in particular«. The described markers have a role in several arterial beds!

 Conclusions, lines 422-426.This paragraph is a long sentence without a verb and does not make sense. Please, rephrase.

 Conclusions, lines 427-430. The last two paragraphs are authors' opinion which has no basis in the review and should be omitted.

 The text needs correction of several typographical errors.

Comments on the Quality of English Language

 The text needs correction of several typographical errors.

Author Response

Thank you for reviewing our paper. Please find attached our responses. 

Reviewer 3 Report

Comments and Suggestions for Authors

Dear Authors, 

The manuscript entitled "Narrative Review of Biological Markers in Chronic Limb Threatening Ischemia" is a very interesting article in the field focused on the finding of biomarkers that can be used  in chronic limb ischemia. Minor revisions are required before the manuscript can be furhter processed.

1. The authors have successfully performed a systemic review using the PRISMA guidelines. 

2. It would be very good for the readers, methods for the identification of the biomarkers (e.g. protemic approaches, flow cytometric analysis) to be included in the manuscript.

3. Are there any evidence for possible association of specific genes with the performed ischemia? 

4. Are there any molecular markers such as the identification of specific genes to be associated with the condition?

5. The individuals HLA or inborn errors of immunity, do they play any particular role in the chronic limb ischemia.

Author Response

(The authors gave the same response as above.)

Round 2

Reviewer 1 Report

Comments and Suggestions for Authors

The presented review paper: 'Narrative Review of Biological Markers in Chronic Limb Threatening Ischemia' is interesting and well-written. The Authors have addressed all my comments in satisfactory way. I have no further suggestions.

I would like to congratulate Authors on their review

Author Response

Thank you for your comments and for evaluating our paper.

Sincerely yours,

Andreea Rata

Email: rataandreealuciana@gmail.com

Reviewer 2 Report

Comments and Suggestions for Authors

Please, provide references for the introductory senteneces of the  biomarkers: IL-6, IL-8, calprotectin.

The discussion contains numerous statements that are not original. Please, provide the appropriate references.

Comments on the Quality of English Language

-

Author Response

Thank you for your comments. We provided references you required.

Sincerely yours,

Andreea Rata

Email: rataandreealuciana@gmail.com